There are amendments to this paper

# Long-range spatio-temporal correlations in multimode fibers for pulse delivery

Wen Xiong[1], Chia Wei Hsu [1] & Hui Cao [1]

Long-range correlations play an essential role in wave transport through disordered media, but have rarely been studied in other complex systems. Here we discover spatio-temporal intensity correlations for an optical pulse propagating through a multimode fiber with strong random mode coupling. Positive long-range correlation arises from multiple scattering in fiber mode space and depends on the statistical distribution of arrival times. By optimizing the incident wavefront of a pulse, we maximize the power transmitted at a selected time, and such control is significantly enhanced by the long-range spatio-temporal correlation. We provide an explicit relation between the correlation and the power enhancement, which agrees with experimental results. Our work shows that multimode fibers provide a fertile ground for studying complex wave phenomena. The strong spatio-temporal correlation can be employed for efficient power delivery at a well-defined time.

[1] Department of Applied Physics, Yale University, New Haven, CT 06520, USA. Correspondence and requests for materials should be addressed to H.C. (email: hui.cao@yale.edu)

Coherent transport of classical and quantum waves in disordered media exhibits long-range correlations, which exist in space, angle, frequency, time, and polarization[1–15]. Such correlations, resulting from the crossing of wave paths, are responsible for the formation of highly transmitting channels in diffusive systems[16–19]. In the frequency domain, long-range correlations enable broad-band enhancement of transmission through disordered media by wavefront shaping[20]. Spatially, long-range correlations significantly increase the efficiency of wave focusing to a target of size much larger than the wavelength in strongly scattering media[21]. However, long-range correlations also increase the background when optimizing the energy delivered to a single speckle grain for continuous waves[22,23] and pulses[24].

From the aspect of scattering, a multimode fiber (MMF) with strong mode mixing shares similarities with a disordered medium. Inherent imperfections and environmental perturbations introduce random mode coupling in an MMF, and its effect grows with the length of the fiber[25,26]. Such coupling can be regarded as scattering in the fiber mode space, leading to energy transfer from the input mode to the other transverse modes. An MMF has a significant difference from the disordered medium: negligible reflection and low propagation loss leading to near-unity transmission. For a continuous wave input, energy conservation dictates that the intensity increase in one mode must be accompanied by intensity decreases in other modes, resulting in negative correlation among the transmitted spatial modes, similar to those found in weak-scattering (ballistic) systems and chaotic cavities[8,27,28]. If the MMF has a large number of modes, such static correlations are very weak. When the input is a short pulse, however, energy is no longer conserved at any particular time, and correlation may be modified and become time-dependent. Nevertheless, little is known about such dynamic correlation in MMFs.

In this work, we discover long-range spatio-temporal correlations in MMFs with strong random mode mixing. For a short pulse input, the transmitted intensities in different spatial channels are generally positively correlated at a given arrival time. The correlation is enhanced at arrival times away from the center of the transmitted pulse, which we attribute to the reduced number of propagation paths at early or late arrival times. The transmitted powers at different delay times are positively correlated for short separation of the delays, and become negatively correlated for distant delays. Such dynamic correlations in an MMF are distinct from those in diffusive or localized random media where the long-range correlations in transmission are always positive[9,24]. The spatio-temporal correlations play a crucial role in the coherent control of short pulses transmitting through an MMF. The positive correlation among spatial channels enables a global enhancement of transmitted energy at a selected arrival time by shaping the incident wavefront. Experimentally, we achieve a higher enhancement when the target time is before or after the mean arrival time, as a result of stronger long-range correlation. Theoretically, we provide a quantitative relation between spatio-temporal correlations and the time-dependent enhancement of transmitted power, which agrees well with our experimental data. Our results show that the maximal power that can be delivered through an MMF at a well-defined time is much higher than what is achievable without long-range correlations. This discovery is important to MMF applications such as telecommunication[29], fluorescence endoscopy[30–32], nonlinear microscopy[33], and fiber amplifiers[34–36], in which ultrashort pulses are deployed for energy delivery.

## Result

### Static and dynamic correlations.
We start with the known static correlations in disordered media. For a monochromatic wave with any given input wavefront, the intensities $I(\mathbf{r})$ at different output positions $\mathbf{r}$ are correlated[7,37,38]

$$\frac{\langle I(\mathbf{r})I(\mathbf{r}+\Delta\mathbf{r})\rangle}{\langle I(\mathbf{r})\rangle\langle I(\mathbf{r}+\Delta\mathbf{r})\rangle} - 1 = F(\Delta\mathbf{r})\tilde{C}_1 + \tilde{C}_2, \quad (1)$$

where $\langle\ldots\rangle$ denotes the ensemble average. The constant $\tilde{C}_1$ gives the strength of short-range correlation, as the normalized function $F(\Delta\mathbf{r})$ decays to zero at the distance $|\Delta\mathbf{r}|$ larger than the speckle size[39]. The constant $\tilde{C}_2$ represents the long-range correlation that results from path crossings[1,4] and is independent of distance (see more details in Supplementary Note 1).

Even though Eq. (1) originates from wave transport in disordered media, it has been derived mathematically on fully general ground[37,40], with the only assumption of isotropy, namely, all channels are fully mixed and are statistically equivalent (see Supplementary Note 1). An MMF with strong and random mode mixing can also exhibit isotropy and therefore follow Eq. (1). However, for continuous wave, the correlation associated with intensity fluctuation in different spatial modes of an MMF is somewhat trivial. Due to energy conservation, an intensity fluctuation of $\Delta I$ in mode $a$ results in an intensity change of $-\Delta I/(N-1)$ in each of the other $N-1$ modes on average, where $N$ is the number of fiber modes. For $N \gg 1$, $\tilde{C}_2 \approx -1/N$ is small.

Pulsed inputs introduce time dependences and non-trivial magnifications to the correlations in MMFs. We consider correlations of the transmitted intensity $I(\mathbf{r}, t)$ between different output positions $\mathbf{r}$ and $\mathbf{r} + \Delta\mathbf{r}$ at arrival times $t$ an $t'$,

$$C(\Delta\mathbf{r}, t, t') \equiv \frac{\langle I(\mathbf{r},t)I(\mathbf{r}+\Delta\mathbf{r},t')\rangle}{\langle I(\mathbf{r},t)\rangle\langle I(\mathbf{r}+\Delta\mathbf{r},t')\rangle} - 1. \quad (2)$$

In the $t = t'$ case, when $C(\Delta\mathbf{r}, t, t)$ is positive, the transmitted power at time $t$ can be efficiently enhanced by wavefront shaping, as enhancing the intensity at one position will simultaneously enhance the intensities at other positions. If the time-dependent transmission matrix at arrival time $t$ is sufficiently isotropic, we expect the same structure as Eq. (1). When $t \neq t'$, the correlation governs the transmitted intensities at time $t$ when the transmission at time $t'$ is modified by changing the incident wavefront; therefore it is related to the temporal shape of the output pulse when the transmitted power is optimized at a given time.

### Time-dependent transmission matrix.
To characterize such spatio-temporal correlation, we measure the transmission matrix of an MMF with strong mode mixing. We use an off-axis holographic setup schematically shown in Fig. 1a. A spatial light modulator (SLM; Hammamatsu X10468) scans the incident angle of a laser beam (Aglient 81940A) onto the MMF, to excite different spatial modes with horizontal polarization. The plane wave of the reference arm and the light transmitted through the fiber interfere to form fringes on the camera, from which we extract the horizontally polarized transmitted field. We use a one-meter-long 0.22-NA graded-index fiber with a core radius of 50 μm and 84 guided modes per polarization. To introduce strong mode mixing into such a short fiber, we use clamps to create micro-bendings. The path lengths of the two arms are matched so the mean arrival time of the pulse (relative to the reference) is zero. The spectral correlation width of the fiber is 0.20 nm at the wavelength of 1550 nm. We measure the field transmission matrices over a wavelength range of 6.4 nm with the step of 0.04 nm. We then perform a Fourier transform to obtain the time-dependent transmission matrices $u(t)$ relating the incident wavefront $|\psi_{\mathrm{in}}\rangle$ to the transmitted wavefront $|\psi_{\mathrm{out}}(t)\rangle = u(t)|\psi_{\mathrm{in}}\rangle$ at different arrival times $t$, considering a Gaussian transform-limited input pulse centered at wavelength 1550 nm with a full

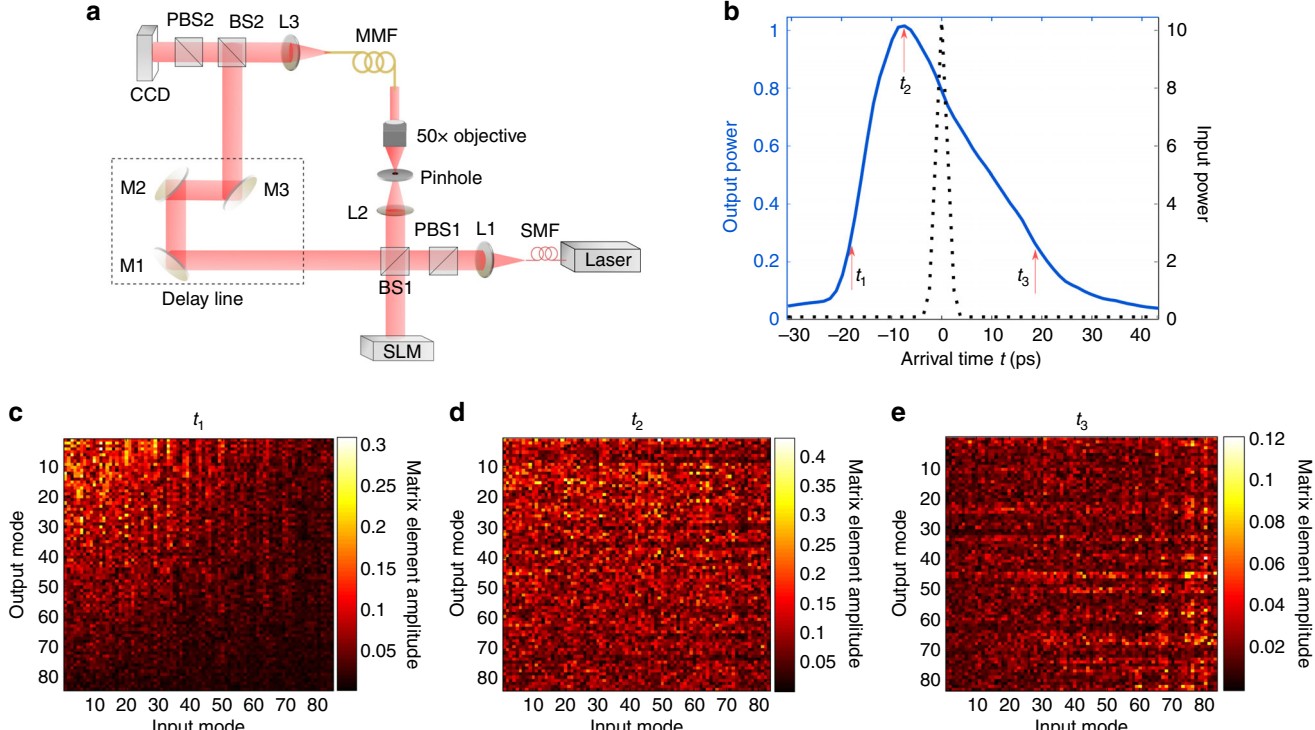

**Fig. 1** Time-dependent transmission matrix of an MMF. **a** Schematics of experimental setup for both transmission matrix measurement and wavefront shaping. A laser beam with tunable frequency is collimated, and its horizontal polarization is selected and split into two arms, with one being the reference and the other propagating through the MMF after reflecting off a spatial light modulator (SLM). The SLM is demagnified and imaged onto the MMF facet. Light transmitted through the MMF is recombined with the reference plane wave, and its horizontal polarization is imaged onto a CCD camera. The path lengths of the two arms are matched by tuning the delay line formed by mirrors M1–M3. L, lens; BS, beam splitter; PBS, polarizing beam splitter. **b** Temporal shapes of the input pulse (black dotted line, right axis) and the mean eigenvalue of $u^{\dagger}(t)u(t)$ (blue solid line, left axis) representing the transmitted intensities of random spatial inputs. The two curves are normalized to have the same area. **c**–**e** Magnitudes of the measured time-dependent transmission matrices at three arrival times (marked by red arrows in **b**), showing strong mode mixing in the fiber. The transmission matrices are measured in **k** space at input and real space **r** at output, and subsequently converted to the fiber mode basis

width at half maximum (FWHM) of 2.0 nm (temporal FWHM = 2.6 ps). The input bandwidth is 10 times of the spectral correlation width of the fiber. Thus for random input wavefronts the transmitted pulse would be 10 times longer than the input pulse.

The total output intensity at time $t$ is $\langle\psi_{\mathrm{out}}(t)|\psi_{\mathrm{out}}(t)\rangle = \langle\psi_{\mathrm{in}}|u^{\dagger}(t)u(t)|\psi_{\mathrm{in}}\rangle$. The mean eigenvalue of $u^{\dagger}(t)u(t)$, shown in Fig. 1b, represents the total transmitted power for random spatial input wavefronts as a function of arrival time $t$. Comparing to the input pulse width (Fig. 1b, black dotted line), the output pulse is significantly stretched and distorted due to strong modal dispersions that different modes propagate at different group delays. The strong random mode coupling is evident from the magnitude of the time-dependent transmission matrix, shown in Fig. 1d for central arrival time; no matter which mode is launched at the input, light is scattered to all spatial modes at the output. The absence of a dominant diagonal reveals negligible ballistic light at fiber output. Higher-order modes have slightly lower magnitudes because they suffer stronger loss than the lower-order modes. The transmission matrix at early (late) arrival time in Fig. 1c (Fig. 1e) has larger contributions from lower-order (higher-order) modes which have shorter (longer) group delay (detailed discussion given in Supplementary Note 1).

**Spatio-temporal correlations**. We calculate the spatio-temporal correlations $C(\Delta\mathbf{r}, t, t')$ from the measured time-dependent transmission matrices, replacing the ensemble average in Eq. (2) with an average over random input spatial profiles. Figure 2a, b

plot $C(\Delta r, t) \equiv C(|\Delta\mathbf{r}|, t, t' = t)$ and two cross sections of it along $\Delta r$ at $t = -17.3$ ps and $t = 3.7$ ps. We observe a short-range correlation that starts from one and vanishes at the speckle size of about 3 μm, beyond which we see a long-range correlation that is approximately constant with respect to $\Delta r$. This indicates $C(\Delta r, t) = F(\Delta r)C_1(t) + C_2(t)$, consistent with Eq. (1). Figure 2c shows the arrival-time dependence of the long-range correlation $C_2(t)$; it is small at the central arrival time but increases toward early or late arrival times.

The time dependence of $C_2(t)$ can be understood through the optical path-length distribution in the multimode fiber. Due to strong mode coupling, there are numerous paths that light can take to travel through the fiber. By exciting the fiber with many random incident wavefronts, all paths are explored and the averaged temporal shape of transmitted pulse in Fig. 1b reflects the number of propagation paths with varying lengths. For the middle delay times, there are a large number of propagation paths, thus $C_2(t)$ is very weak. The larger $C_2(t)$ at early and late arrival times is consistent with the lower number of paths for such times. Conceptually, if there is only one path of length corresponding to the arrival time $t$, the output intensities $I(\mathbf{r}, t)$ at different positions $\mathbf{r}$ must be fully correlated: varying the incident wavefront can only change how much light is coupled into that one path, which will increase or decrease $I(\mathbf{r}, t)$ at all positions in the same way. Therefore, the fewer paths for the arrival time $t$, the stronger $C_2(t)$.

Long-range correlation between far-away speckle grains exists not only between speckle grains at the same arrival time, but also

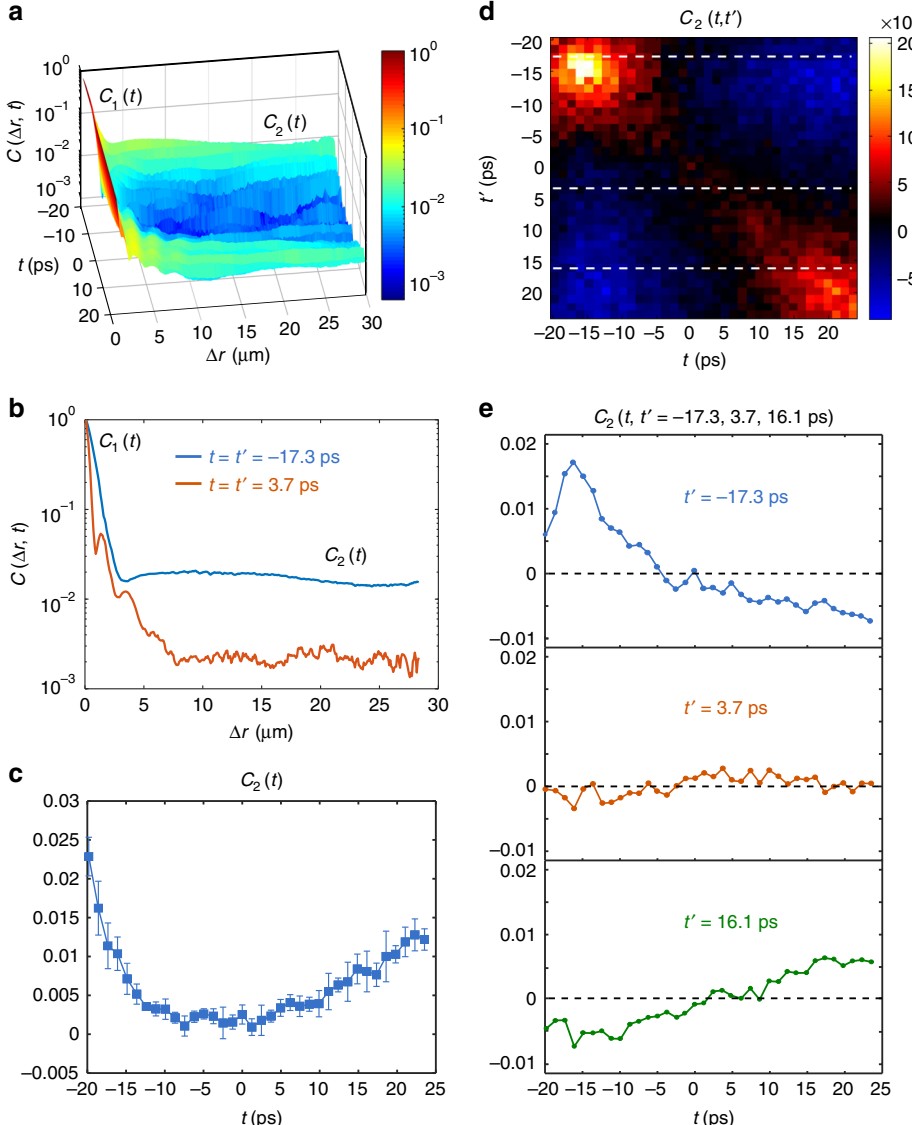

**Fig. 2** Spatio-temporal correlations in MMF with strong random mode mixing. **a** Intensity correlations $C(\Delta r, t) \equiv C(|\Delta \mathbf{r}|, t, t' = t)$, revealing a short-range component $C_1(t) \approx 1$ at spatial distance within one speckle ($\Delta r \lesssim 3\,\mu m$) and a long-range component $C_2(t)$ that persists at large distance. **b** Two cross sections of $C(\Delta r, t)$ at arrival times $t = -17.3$ ps and $t = 3.7$ ps. **c** Time dependence of the long-range component $C_2(t)$, averaging over $\Delta r$ for $\Delta r > 5\,\mu m$. The error bars represent the standard deviation among four measurements of the fiber in different bending configurations. **d** Long-range correlations $C_2(t, t')$ between spatio-temporal speckle grains at different arrival times $t$ and $t'$. **e** Cross sections of $C_2(t, t')$ at $t' = -17.3$ ps, 3.7 ps and 16.1 ps (marked by white dashed lines in **d**)

between speckle grains at different arrival times. This is quantified by $C(\Delta \mathbf{r}, t, t')$ as defined in Eq. (2). At large $|\Delta \mathbf{r}|$, this quantity again becomes independent of $|\Delta \mathbf{r}|$ and approaches the asymptotic value $C_2(t, t')$. In Fig. 2d, we show $C_2(t, t')$ for $t$ and $t'$ from $-20$ ps to 25 ps. The long-range correlation is positive close to the diagonal, namely close to the $C_2(t)$ discussed earlier. When $t$ and $t'$ are far apart, however, the long-range correlation becomes negative. Figure 2e show three cross-sections. Near the central arrival time ($t' = 3.7$ ps), $C_2(t, t')$ is close to zero at all $t$. Meanwhile, at $t' = -17.3$ ps, $C_2(t, t')$ peaks at $t \approx t'$ and decays away from it, eventually becoming negative. The trend, however, is opposite at $t' = 16.1$ ps. The correlation is negative at early delay times and becomes positive at late arrival time. Such a negative correlation is a result of the conservation of transmitted pulse energy, which requires an increase of spatially integrated intensity (power) at arrival time $t = t'$ to be compensated by a decrease of power at other arrival times.

**Pulse delivery.** The positive spatio-temporal correlation $C_2(t)$ at early or late arrival times will lead to a higher achievable enhancement at such times. Because the matrix $u^\dagger(t_0)u(t_0)$ is Hermitian, the global optimum, which determines the maximum power that can be delivered at time $t_0$, is given by the largest eigenvalue of $u^\dagger(t_0)u(t_0)$, and the corresponding eigenvector is the desired incident wavefront[20]. By shaping the incident wavefront with the SLM[41,42], we can enhance the total transmitted power at a target arrival time and compensate for the strong modal dispersion in the fiber. Experimentally, we determine such optimal transmission channels from the measured time-dependent transmission matrices, and then generate the desired wavefront with the same setup, using computer-generated phase holograms to simultaneously modulate the phase and amplitude profiles[43]. By scanning the wavelength and Fourier transforming the spectral measurements to the time domain, we obtain the

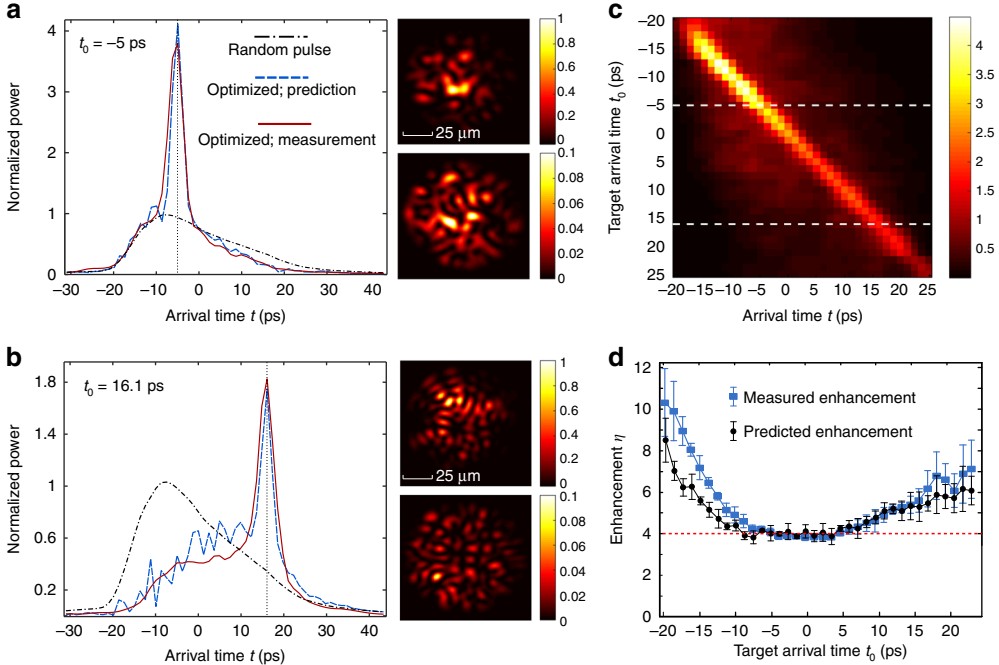

**Fig. 3** Enhancing transmitted power at selected time. **a**, **b** Temporal shapes of the output pulse when the spatially integrated intensity (power) is optimized at arrival time (**a**) $t_0 = -5$ ps and (**b**) $t_0 = 16.1$ ps. Red solid lines are the measured pulse shapes with optimized input wavefronts, the blue dotted lines are predicted from measured correlations $C_2(t, t')$. They exhibit strong enhancement compared to the mean eigenvalues of $u^\dagger(t)u(t)$, which represents the mean output pulse shape for random input wavefronts (black dash-dotted lines). Transmitted powers are all normalized by the peak power of the transmitted pulse with random input wavefronts. The insets are spatial intensity patterns at $t_0$ for the optimized wavefront (upper panel) and a random wavefront (lower panel). The speckle grains at $t_0 = -5$ ps are larger than those at $t_0 = 16.1$ ps, due to larger contributions from the lower-order modes at earlier arrival time. **c** Temporal shapes of pulses optimized at different $t_0$, ranging from $-20$ ps to 25 ps. The target time of **a** and **b** are marked by the white dashed lines. **d** Enhancement factor $\eta$ of the transmitted power at the target arrival time $t_0$. Blue squares: measured enhancement. Black circles: enhancement predicted from $C_2(t)$. Error bars represent the standard deviation among four measurements of the fiber in different bending configurations. Red dashed line indicates four times enhancement if $C_2(t) = 0$

spatially integrated temporal pulse shapes of such optimal transmission channels.

The pulses optimized for arrival times $t_0 = -5$ ps and $t_0 = 16.1$ ps are shown in Fig. 3a, b (red solid curve), in comparison to the averaged pulse of random spatial inputs (black dash-dotted curve). The sharp peak at the selected arrival time, as marked by the vertical black dotted line, illustrates that the transmitted power can be effectively enhanced at different target times, even in the presence of strong modal dispersions in the fiber. The peak width equals the input pulse width. The spatial intensity patterns of the optimized pulse and the non-optimized one at the target arrival times, shown in Fig. 3a, b, are obtained from the Fourier transform of the frequency-resolved field patterns measured with the optimized and random incident wavefronts. It is distinct from spatio-temporal focusing[24,44–48] where only one speckle is enhanced. Figure 3b shows that the transmitted power after the target time increases, but well before the target time it increases. Such changes are determined by correlation $C_2(t, t' = 16.1$ ps) shown in Fig. 2e. The negative correlation at early arrival time suppresses the background and the positive correlation at late arrival time enhances the background. Figure 3c plots the pulse shapes optimized for different arrival times from $t_0 = -20$ ps to 25 ps. The peak follows the target time $t_0$, and notably, the background also shifts with the target time.

To evaluate the effectiveness of the transmitted power optimization, we define an enhancement factor $\eta(t_0) \equiv I_{enh}(t_0)/I_{random}(t_0)$, where $I_{enh}(t_0)$ and $I_{random}(t_0)$ are the spatially integrated intensities of the optimized pulse and the random pulse at the target time $t_0$. We plot the measured enhancement

factor $\eta$ (blue square) in Fig. 3d. The standard deviation of the enhancement between measurements on four different days is shown by the error bars. The deviation is larger at early or late arrival times as the weaker pulse intensities there lead to smaller signal-to-noise ratio. The average enhancement is about four times around the central arrival time, which is what one expects through the quarter-circle law for the singular values of a square random matrix with uncorrelated elements[49]. At early or late arrival times, we achieve power enhancements much >4; such increase is consistent with the large long-range correlation $C_2(t)$ that we observed (Fig. 2c).

**Power enhancement.** Finally, we provide a quantitative connection between the long-range spatio-temporal correlation $C_2(t)$ and the enhancement factor $\eta(t_0)$ of transmitted power at arrival time $t_0$. We use a heuristic model similar to that employed in ref. [21], capturing the correlation between output channels at arrival time $t_0$ through a reduction in the effective number of output channels. Specifically, we consider an effective random matrix with $N$ input channels and $N^{(eff)}(t_0)$ output channels, and we consider all elements of this matrix to be identically independently distributed. The enhancement $\eta$ is determined by the largest eigenvalue, which is related to the spread of the eigenvalues characterized by the eigenvalue variance. As detailed in the Supplementary Note 1, the normalized eigenvalue variance associated with the reduced matrix is given by the Marčhenko–Pastur distribution[49] to be $N/N^{(eff)}(t_0)$, while that associated with the actual time-resolved transmission matrix is

$1 + NC_2(t_0)$. Therefore, we choose

$$N^{(\text{eff})}(t_0) = \frac{N}{1 + NC_2(t_0)} \qquad (3)$$

to match the two corresponding eigenvalue variances. This relation quantifies how long-range correlation effectively reduces the number of output channels. The enhancement is the normalized maximal eigenvalue, which for an uncorrelated matrix is $\tau_{\max}/\bar{\tau} = \left(1 + \sqrt{N/N^{(\text{eff})}(t_0)}\right)^2$ (ref. [49]). Inserting Eq. (3), we obtain a simple equation

$$\eta(t_0) = \left(1 + \sqrt{1 + NC_2(t_0)}\right)^2 \qquad (4)$$

that relates the maximal enhancement to the long-range spatio-temporal correlation.

In Fig. 3d, we compare the measured enhancement to the enhancement predicted through the measured $C_2(t)$ via Eq. (4) (black circles). Overall, the two curves agree well, especially around the central arrival time. Some differences at early or late arrival times may be due to the fact that the time-dependent transmission matrix is not as isotropic as that at the central time (as shown in Fig. 3c–e). We further generalize the relationship in Eq. (4) to predict the whole output pulse (both the peak at the target time and the background) via $C_2(t, t')$ (see Supplementary Note 1). In Fig. 3a, b, we plot the predicted temporal shapes (blue dotted curves) of the optimized pulses with $t_0 = -5$ ps and $t_0 = 16.1$ ps on top of the measured pulse shapes. As $C_2(t, t_0)$ changes from positive correlation for the arrival time $t$ close to the target time $t_0$ to negative correlation for $t$ far from $t_0$, the transmitted power is enhanced near the peak at $t_0$ and suppressed away from the peak. Consequently, the background shifts toward the peak due to long-range correlation.

## Discussion

Local and nonlocal correlations have been studied extensively in scattering media, but there are few observations in other complex photonic systems. Short-range correlation introduces the rotational memory effect that has been observed in a MMF with weak mode coupling[50,51]. Here we observe long-range spatio-temporal correlation in a multimode fiber with strong mode mixing when a pulse propagates through the fiber. The correlation not only determines the effectiveness of enhancing the transmitted power at a target time, but also capture the temporal shape of the resulting pulse. We provide a qualitative explanation for the long-range spatio-temporal correlations using the optical path-length distribution in the multimode fiber with random mode mixing. This simple model reveals the possibility of physically turning the spatio-temporal correlations by tailoring the path-length distribution in the fiber via a careful design of the fiber configuration.

Enhancing the transmitted power in time can be utilized in many fiber applications from communication to imaging. The maximum eigenmode (EM) of the time-resolved transmission matrix provides the incident wavefront for focusing the transmitted pulse to a chosen delay time. This method is effective for any input pulse with arbitrarily broad spectrum, and it guarantees the maximal power delivery at any selected time. Especially when the spectral width of an input pulse is much larger than the spectral correlation width of the fiber, the EM outperforms the principal mode[52–54] and super-principal mode[55] in achieving the highest peak power of the transmitted pulse (see Supplementary Note 2 for detailed discussion and direct comparison).

## Data availability

The data that support the findings of this study are available from the corresponding author upon reasonable request.

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

## Acknowledgements
We acknowledge Stefan Rotter, Hasan Yilmaz, Tsampikos Kottos, Douglas Stone, and Alexandre Aubry for stimulating discussions. This work is supported by US National Science Foundation under the Grant No. ECS-1809099.

## Author contributions
H.C. initiated and led the study. W.X. performed the experiment and numerical simulation. C.W.H. provided analytical derivation. All authors contributed to data analysis and the manuscript.

## Additional information

**Competing interests:** The authors declare no competing interests.

