## [Peer Review File · Nature Communications]

Reviewers' comments:

Reviewer #1 (Remarks to the Author):

In the paper authors investigate experimentally spatio-temporal correlations in a pulse that has propagated through a multimode optical fiber. They observe similarities with behavior of disordered media. They also measure the time-dependent transmission matrix use an SLM to shape the input pulse such that the output pulse is sharply localized in time.

The research presented in the paper is interesting and it is valid as far as I can judge. However, I do not think that the results are interesting enough to guarantee publication in Nature Communications, which I therefore cannot recommend. I would recommend the paper for some less impacted journal.

Reviewer #2 (Remarks to the Author):

This paper shows that even though steady-state transmission through a perturbed optical fiber is nearly unity for any incident wave, the intensity of pulsed transmission through the fiber can still be controlled by manipulating the incident wavefront. Coherent measurements are made of the time varying transmission matrix through a fiber and of short- and long-range intensity correlation across the fiber cross section. Measurements of optimized pulse intensity at different delay times are consistent with a simple expression related to the degree of intensity correlation.

This work is closely related to studies of enhanced pulsed transmission through random media. One paper, by Shi et al. in OL 38, 2714 (2013), deals with spatiotemporal control of microwave radiation propagating through a random waveguide. It shows the enhancement of transmitted intensity at selected times in a pulse and the variation of the enhancement with the participation number of transmission eigenvalues. This quantity is essentially the inverse of the degree of long-range correlation and gives the contrast between the peak intensity at a selected point and the background. The pulse enhancement varies with time delay because of the variation of the effective number of transmission eigenchannel contributing to transmission. The figures in the two papers are so similar that it would be worth looking for points of similarity and difference.

The work should also be distinguished from previous work by some of the authors. In particular, the differences between the present work and Ref. 48, by Xiong, et al. on Spatiotemporal control of light transmission through a multimode fiber with strong mode coupling in PRL 117, 053901 (2016), should be spelled out.

This paper presents elegant measurements of intensity correlation and optimal enhancement in a pulse transmitted through an optical fiber. The results are well explained by the theory presented. Since optical fibers are important delivery systems for light in laboratory experiments and in telecommunications and sensing, there are likely to be many applications for the tailored pulses demonstrated here. I therefore would recommend the paper for publication in Nature Communications once the points mentioned above are considered.

A few minor points:

-The intensity patterns for random incident wave fronts in the lower insets for Figs. 3a,b can't be made out so should have a different color bar for lower panels where the intensity is weaker.

-“Such long-range dynamic correlations in an MMF are distinct from and can be much stronger than those predicted to exist in a random scattering medium [9].” But the same effect is reported in the paper mentioned about by Shi et. al which describes correlation of pulsed intensity through a

random medium.

-“the output pulse is significantly broadened due to mixing between modes that propagate at different group delays.” But usually it is the mixing of modes in an optical fiber that tends to narrow the pulse, something like motional narrowing.

-“Higher-order modes have slightly lower magnitudes due to mode-dependent loss in the fiber.” But if modes are mixed there should be little difference in delay time for different incident modes.

Some grammatical points:

-speckle does not have a plural and it is usually an adjective. So the phrase “within one speckle” should be “within one speckle spot” and “between spatio-temporal speckles” should be “between spatio-temporal speckle”.

-“correlation” should be singular unless one means specifically to point out different types of correlation, So it should be “Long-range correlation” not “Long-range correlations”

-:Here we discover” could be “Here we report the discovery of

-“We provide an explicit relation between the correlations and the enhancements, which closely agrees with experimental data.” Could be “We provide an explicit relation between the correlation and enhancement of intensity, which agrees with experimental results.”

-“from crossings of wave paths” could be “from the crossing of wave paths”

-“conversed” should be “conserved”

-“and correlations may be modified and become time-dependent” there is no change in the nature of correlation, it is just that in incident intensity varies in time so the correlation measured will also vary in time.

-“agrees well to” could be “agrees well with”

-“denotes ensemble average” should be “denotes the ensemble average”

-“an one-meter-long” should be “a one-meter-long”

-“...introduce strong mixing to” should be “...introduce strong mixing into”

-“to time domain” should be “to the time domain”

Reviewer #3 (Remarks to the Author):

Spatio-temporal correlations in multimode fibers for pulse delivery

In this paper, the authors are experimentally measuring the transmission matrix with spectral resolution of a step index multimode fibre over a narrow bandwidth (6nm) in the strong mode mixing regime. With a Fourier transform they obtain the transmission matrix in the delay domain. From this set of data, they show for the first time the existence of spatio temporal correlations of second order in a multimode fibre. They show how these positive correlations could be useful to enhance light delivery in the time domain. They also provide numerical results to support the experimental data.

The study of statistical correlations in disordered media has been experimentally studied over the last 30 years. Most of the study were focused on multiple scattering systems, in open scattering samples. The authors are here presenting the existence of such correlation in multimode fibre, which could be interesting for the optics community.

The paper is easy to read, even though sometimes the authors are assuming hypothesis that might not be clear for the non expert reader and would deserve more clarity.

My comments and questions are the following:

- All the results of the correlations are based on the fact that the transmission matrix of the fibre with time resolution is "isotropic" (left column of page 2 of the manuscript). However, in the paper and in the supplementary material, the authors never demonstrate this hypothesis. How could you do it? Maybe the author should show the singular value decomposition of the three matrices of Fig. 1, and that it follows the Marcenko Pastur law? Would that be enough? Is the mode dependent loss of the fibre affecting the "isotropic" criteria?

- The spatial C2 correlation in scattering media has been with wavefront shaping (Vellekoop and Mosk, Universal Optimal Transmission of Light Through Disordered Materials PRL 2008). Upon focusing, the total transmitted energy is increasing via the C2 correlations. If we try to find an analogy with the C2 correlations in multimode fibre, would we see a similar effect? (i.e. the background at different times than t_0 should be enhanced as well?). It seems not to be the case in Fig. 3c, or the effect is maybe too small to be observed. Hence it is not clear that the temporal shape of the achieved output pulse is only ruled by the measured so-called "spatio-temporal correlations".

- In this paper the authors are only dealing with the long range correlations, and not about the C1 and the C3. I would suggest to modify slightly the title, which suggests the authors are studying more than the C2 effect. I would also recommend the author to cite this paper Amitonova et al., "Rotational memory effect of a multimode fiber", Optics express 2015 where the rotational memory effect is demonstrated experimental in a MMF, as the angular memory effect cannot exist in a MMF.

- In page 2 left column, the authors are stating "the correlations associated with an MMF is somewhat trivial, as $C2 \sim 1/N$ is small". Maybe the authors should explain a bit more, it might not be trivial to the non-expert reader.

- Regarding the experimental measurements, how long does it take to scan the wavelength for a fixed input state (end of page 4, right column)? Is the phase of the laser stable enough?

- In page 3 left column, "considering a Gaussian transform-limited input pulse centered at wavelength 1550 nm with a full width at half maximum (FWHM) of 2.0 nm (temporal FWHM = 2.6 ps)". How do you experimentally consider this when you measure your data, as you perform a monochromatic scan of the wavelength? You are not really sending an ultrashort pulse in your experiment. This point should be clarified. Why did you consider these numbers and not a broader source for instance? Is that limited by the resolution of your measurements?

- In Fig. 3b, why is the speckle size changing according to the delay time?

- How did you measure the error bars in Fig.3d? Did you try with the same fibre but different bending of the fibre? How many measurements did you perform to measure the error bars?

- In Fig. 4, what is defining and limiting the width of the temporal peak? Could it be narrower? (as you perform a spectral scan and you don't have any limitation of the time width of an ultrashort pulse) How do you normalize the intensities in Fig.4a and Fig.4b? How do you measure the speckle with the spatial resolution at a chosen time? Is there a reason why light is mostly focused on two speckle grains in Fig. 4a and Fig.4b when you enhance pulse delivery at a chosen delay?

- In page 4 left column "the large $C2(t)$ at early and late arrival times is consistent with the lower number of paths at such times", maybe the author should plot the number of modes at such times, to make it more clear? Could this be quantitative?

- This sentence in page 5 left column "The background change is a result of long-range spatio-temporal

correlations $C_2(t; t_0)$. Namely, the negative correlations for well-separated times t and t_0 ($t_0 = t_0$) shift the background toward the target time t_0 is not clear

- In page 5 left column, "The average enhancement is about 4 times around the central arrival time, which is what one expects through the quarter-circle law for the singular values of a square random matrix with uncorrelated elements". Don't you have any ballistic light in multimode fibres, even in the strong mode mixing regime? An evidence of this would be highly recommended.

- Are the measured C_2 correlations the only parameters to help building up the pulse delivery? In the case of the spatio-temporal focusing in multimode fibre (Morales-Delgado et al. Optics Express Vol. 23, Issue 7, pp. 9109-9120 (2015)), should the C_2 correlation play a role as well in the digital phase conjugation results? If you were not in a strong mode-mixing regime, would you expect similar results, slightly better or worse?

- In page 5 right column. "For the broad-band pulses studied here", this sentence is incorrect, as you are not really sending a short pulse in the experimental results.

- In the Supplementary information part C, what is the delay Δt over which $C_2(\Delta t)$ goes to 0?

- My final question is about your comparison between the achieved state and the super principal modes. Is there a physical relationship between the principal modes and the EM? For instance, the bandwidth in the spectral correlation when you calculate the principal modes is at 0.9. If you would like to optimize the bandwidth at 0.6 (which seems to be where the bandwidth of the EM modes are the broadest) instead of 0.9, could you generate an equivalent of the EM based on the principal modes measurement only?

Side remarks:

- In Fig. 1b, there must be a problem of unit of the input power if the energy is conserved (the max must be 0.4 and not 0.04?)

- What is the physical meaning of $N^{\text{eff}}(t_0)$?

If the authors could clarify those points it would be highly appreciated.

Reply to Reviewer #1:

In the paper authors investigate experimentally spatio-temporal correlations in a pulse that has propagated through a multimode optical fiber. They observe similarities with behavior of disordered media. They also measure the time-dependent transmission matrix use an SLM to shape the input pulse such that the output pulse is sharply localized in time.

The research presented in the paper is interesting and it is valid as far as I can judge. However, I do not think that the results are interesting enough to guarantee publication in Nature Communications, which I therefore cannot recommend. I would recommend the paper for some less impacted journal.

We wish to stress the significance and interest of this work to both fundamental physics and practical application.

From the fundamental physics point of view, we not only demonstrate the existence of spatio-temporal correlations of second order in multimode fibers, but also illustrate their qualitative difference from the long-range correlations in diffusive random media. For example, the second-order correlation $C_2(t, t')$ is positive for small $|t-t'|$ and turns negative for large $|t-t'|$ in a multimode fiber with random mode mixing, but in a diffusive random medium the second-order correlations are always positive for transmission. These results reveal the fundamental difference between the multimode fiber and the diffusive random medium. In a fiber there is no reflection and when the loss is negligible the energy conservation in transmission result in negative long-range correlation; but in a diffusive random medium, the strong reflection removes energy conservation for transmission.

From the application point of view, the long-range dynamic correlation dictates short pulse delivery through a multimode fiber. The enhancement of transmitted power at a target arrival time t_0 is determined by $C_2(t_0, t_0)$, and the power change at any other time t is given by $C_2(t, t_0)$. This work provides a general method for finding the incident wavefront for focusing the transmitted pulse in time. It is effective for pulses with arbitrarily broad spectra and guarantees the maximal power delivered at any arrival time. For broad-band (short) pulses, this method outperforms the principal modes and super principal modes in achieving the highest peak power of the transmitted pulse. Since pulse delivery is critical to application of multimode fibers in optical communication, imaging and sensing, this work is of practical importance.

Reply to Reviewer #2:

This paper shows that even though steady-state transmission through a perturbed optical fiber is nearly unity for any incident wave, the intensity of pulsed transmission through the fiber can still be controlled by manipulating the incident wavefront. Coherent measurements are made of the time varying transmission matrix through a fiber and of short- and long-range intensity correlation across the fiber cross section. Measurements of optimized pulse intensity at different delay times are consistent with a simple expression related to the degree of intensity correlation.

We appreciate the referee's insightful comments and valuable suggestions that have helped us to improve the manuscript.

1. This work is closely related to studies of enhanced pulsed transmission through random media. One paper, by Shi et al. in OL 38, 2714 (2013), deals with spatiotemporal control of microwave radiation propagating through a random waveguide. It shows the enhancement of transmitted intensity at selected times in a pulse and the variation of the enhancement with the participation number of transmission eigenvalues. This quantity is essentially the inverse of the degree of long-range correlation and gives the contrast between the peak intensity at a selected point and the background. The pulse enhancement varies with time delay because of the variation of the effective number of transmission eigenchannel contributing to transmission. The figures in the two papers are so similar that it would be worth looking for points of similarity and difference.

We thank the referee for bringing up the paper by Shi et al, and we compare it to our paper below. Shi et al studied spatio-temporal focusing (namely, enhancement of intensity at a single speckle in space and time), while we consider the enhancement of all spatial channels at the target time. Shi et al examined the peak-to-background contrast (ratio of the intensity of the speckle at the focus to the intensity of a speckle away from the focus), while we study the enhancement of transmitted power (sum of intensities of all speckles) at a target arrival time. In the paper of Shi et al, the definition of long-range correlation is not explicitly given, but the referee is correct that the eigenvalue participation number M in that paper is essentially the inverse of the long-range correlation C_2 . Therefore, the work of Shi et al and ours uncover different consequences of long-range correlations in complex media.

In the revised manuscript, we have cited the paper of Shi et al and explain the difference of this work from that of Shi et al by adding the following sentence to the first paragraph of our introduction:

“However, long-range correlations also increase the background when optimizing the energy delivered to a single speckle grain for continuous waves [22, 23] and pulses [24].”

2. The work should also be distinguished from previous work by some of the authors. In particular, the differences between the present work and Ref. 48, by Xiong, et al. on Spatiotemporal control of light transmission through a multimode fiber with strong mode coupling in PRL 117, 053901 (2016), should be spelled out.

This work is substantially different from our previous studies on principal modes (PMs) and super-principal modes (super-PMs) in Refs. 53-55.

First, a main discovery of this work is the spatio-temporal long-range correlations in multimode fibers and their consequences, to the best of our knowledge, have never been found before, and certainly not in our previous papers.

Second, this work and our previous studies on PMs and super-PMs consider different types of spatiotemporal controls. Here, we aim to maximize the transmitted power (spatially integrated intensity) at a target arrival time and do not care whether or how the output spatial field profile changes with time or frequency. PMs and super-PMs, on the other hand, aim to achieve an output spatial profile that is invariant in time so that the output field pattern does not change with the arrival time and maintains the spatial coherence; however, the transmitted power is not necessarily maximized at any arrival time. The two types of control aim to optimize different figures of merit.

There are indeed connections between the maximal eigenchannel here and the PMs and super-PMs. In the revised Section II of the Supplementary Materials, we show in Fig. S2 that when the spectral bandwidth of an input pulse is comparable to the spectral correlation width of the multimode fiber, the maximal transmission eigenchannel and the PM/super-PM achieve the same performance in terms of suppressing spectral decorrelation and minimizing temporal distortion of the transmitted pulse. However, when the input has a bandwidth much larger than the spectral correlation width of the fiber, the maximal transmission eigenchannel is the most effective in maximizing the peak power of the transmitted pulse.

In addition to the Supplementary Materials, we also modified the last paragraph of the manuscript to emphasize these points.

This paper presents elegant measurements of intensity correlation and optimal enhancement in a pulse transmitted through an optical fiber. The results are well explained by the theory presented. Since optical fibers are important delivery systems for light in laboratory experiments and in telecommunications and sensing, there are likely to be many applications for the tailored pulses demonstrated here. I therefore would recommend the paper for publication in Nature Communications once the points mentioned above are considered.

We thank Reviewer #2 for the positive recommendation.

A few minor points:

3. The intensity patterns for random incident wave fronts in the lower insets for Figs. 3a,b can't be made out so should have a different color bar for lower panels where the intensity is weaker.

We have changed the color bar accordingly.

4. "Such long-range dynamic correlations in an MMF are distinct from and can be much stronger than those predicted to exist in a random scattering medium [9]." But the same effect is reported in the paper mentioned about by Shi et. al which describes correlation of pulsed intensity through a random medium.

The quoted sentence in the original manuscript referred to random scattering media in wide slab geometries with open boundaries, where the number of spatial modes is large, and the long-range correlation is weak. But as the referee pointed out, in quasi-1D geometries with closed boundaries as studied by Shi et al, the number of modes is small, and the dimensionless conductance is low, thus the long-range correlations become substantial. Therefore, we

modified the sentence to highlight the qualitative difference in the long-range correlation between the multimode fiber and the random scattering medium.

To show the effect we reported here is not the same as that by Shi et al, we compare the inverse of the long-range correlation $C_2(t,t)$, which is equal to the eigenvalue participation number M in the paper of Shi et al. In the multimode fiber with strong mode coupling and negligible loss (left panel), M is symmetric with respect to the mean arrival time, while in the random medium, M is asymmetric (right panel, from Shi et al.).

[Redacted]

Fig. R1. Participation number M of the transmission eigenvalues for the time-resolved transmission matrix of a multimode fiber supporting 86 modes with strong mode coupling (left panel) and quasi-1D disordered media (right panel, copied from Shi et al, OL 38, 2714 (2013))

In addition to the quantitative difference of $C_2(t,t)$, $C_2(t,t')$ displays a qualitative difference between the multimode fiber and the disordered media. As shown in Fig. 2, $C_2(t,t')$ changes from positive correlation at small separation of t and t' to negative correlation at large separation. But in diffusive and localized random media, the long-range correlations are always positive for transmission.

In the revised manuscript, we replace the quoted sentence by the following one:

“The transmitted powers at different delay times are positively correlated for short separation of the delays and become negatively correlated for distant delays. Such dynamic correlations in the MMFs are distinct from those in diffusive or localized random media where the long-range correlations in transmission are always positive [9, 24].”

5. “the output pulse is significantly broadened due to mixing between modes that propagate at different group delays.” But usually it is the mixing of modes in an optical fiber that tends to narrow the pulse, something like motional narrowing.

If there is no mode mixing in a fiber, a pulse launched to a single fiber mode will stay in that mode when propagating through the fiber, thus the output pulse is as short as the input pulse. If there is mode mixing, part of the pulse will be coupled to other modes of the fiber, and the transmitted pulse will be broadened due to modal dispersion. But if the incident pulse is injected to all fiber modes, the pulse width will increase linearly with the fiber length L in the absence of

mode mixing, with mode mixing the pulse width will increase as $L^{0.5}$. The slowing down of pulse broadening by mode mixing is like motional narrowing. Regardless of the launching condition, the output pulse is always broader than the input pulse in the presence of mode mixing.

For clarification, we correct the quoted sentence to

“Comparing to the input pulse width (Fig. 1b, black dotted line), *the output pulse is significantly broadened due to strong modal dispersions that different modes propagate at different group delays*”.

6. “Higher-order modes have slightly lower magnitudes due to mode-dependent loss in the fiber.” But if modes are mixed there should be little difference in delay time for different incident modes.

The quoted sentence is in the discussion of the field transmission matrix for the central (mean) arrival time (Fig. 1d). The distribution of transmitted intensities in all fiber modes depends on the relative strength of mode mixing and mode-dependent loss. Since mode mixing can be regarded as scattering in fiber mode space, we define the transport mean free path l_t as the propagation length at which light originally injected to a single mode is scattered to all modes. If the fiber length L is much larger than l_t and l_t is much smaller than the absorption length l_a for all fiber modes, mode mixing dominates over loss, then the higher order modes should have similar magnitudes to the lower order modes. But these conditions are not met in our fibers. The higher order modes experience more loss and have shorter l_a than the lower order modes. As their l_a become shorter than l_t , the higher order modes will dissipate faster than mode mixing, and their magnitudes would be smaller than those of the lower order modes.

But the referee’s question seems related to the transmission matrix at different delay times. Even if the mode-dependent loss is negligible in the fiber, for short delay time, light must travel mostly in the lower order modes from the beginning to the end of the fiber, so the lower-order modes have more contributions to the transmission matrix of short delay. The shorter the delay, the larger the contributions of the lower-order modes. This is related to the isotropy of the time-dependent transmission matrix.

We have added a subsection to the Supplementary Materials, Section I. D, on the isotropy of transmission matrix.

7. Some grammatical points:

-speckle does not have a plural and it is usually an adjective. So the phrase “within one speckle” should be “within one speckle spot” and “between spatio-temporal speckles” should be “between spatio-temporal speckle”.

-“correlation” should be singular unless one means specifically to point out different types of correlation, So it should be “Long-range correlation” not “Long-range correlations”

-:Here we discover” could be “Here we report the discovery of

-“We provide an explicit relation between the correlations and the enhancements, which closely agrees with experimental data.” Could be “We provide an explicit relation between the correlation and enhancement of intensity, which agrees with experimental results.”

-“from crossings of wave paths” could be “from the crossing of wave paths”

-“conversed” should be “conserved”

-“and correlations may be modified and become time-dependent” there is no change in the

nature of correlation, it is just that in incident intensity varies in time so the correlation measured will also vary in time.

-“agrees well to” could be “agrees well with”

-“denotes ensemble average” should be “denotes the ensemble average”

-“an one-meter-long” should be “a one-meter-long”

-“...introduce strong mixing to” should be “...introduce strong mixing into”

-“to time domain” should be “to the time domain”

We thank the reviewer for reading the manuscript thoroughly and carefully. We have implemented the grammatical corrections in the revised manuscript. We changed to correlation (singular) except when referring to multiple types of correlation.

Reply to Reviewer #3:

In this paper, the authors are experimentally measuring the transmission matrix with spectral resolution of a step index multimode fibre over a narrow bandwidth (6nm) in the strong mode mixing regime. With a Fourier transform they obtain the transmission matrix in the delay domain. From this set of data, they show for the first time the existence of spatio temporal correlations of second order in a multimode fibre. They show how these positive correlations could be useful to enhance light delivery in the time domain. They also provide numerical results to support the experimental data.

The study of statistical correlations in disordered media has been experimentally studied over the last 30 years. Most of the study were focused on multiple scattering systems, in open scattering samples. The authors are here presenting the existence of such correlation in multimode fibre, which could be interesting for the optics community.

The paper is easy to read, even though sometimes the authors are assuming hypothesis that might not be clear for the non expert reader and would deserve more clarity.

We thank the referee for detailed and constructive comments. We have revised the manuscript based on the referee's suggestions.

My comments and questions are the following:

1. All the results of the correlations are based on the fact that the transmission matrix of the fibre with time resolution is "isotropic" (left column of page 2 of the manuscript). However, in the paper and in the supplementary material, the authors never demonstrate this hypothesis. How could you do it? Maybe the author should show the singular value decomposition of the three matrices of Fig. 1, and that it follows the marcenko pastur law? Would that be enough? Is the mode dependant loss of the fibre affecting the "isotropic" criteria?

Let us first clarify that the isotropy of a matrix is not related to the Marchenko-Pastur law. Marchenko-Pastur law states that when elements of a random matrix are uncorrelated to each other, the singular values of the matrix follow certain distributions (a quarter circle in the case of a square matrix). Matrices with correlated elements can be isotropic but do not follow the Marchenko-Pastur law. We show that the time-dependent transmission matrices in the multimode fiber exhibit correlations except at the central arrival time (negligible correlation for large N), they are not expected to follow the Marchenko-Pastur law.

The "isotropy" means that all channels are fully mixed and are statistically equivalent. For the transmission matrix of the multimode fiber, "isotropy" means no matter which mode the incident light is launched into, the transmitted light is uniformly distributed over all spatial modes. This is clearly an approximation for the time-dependent transmission matrix, especially at very early or late delay times.

To check how good the approximation is, we compute the participation ratio $PR = (\sum_{i=1}^N I_i)^2 / N \sum_{i=1}^N I_i^2$, where I_i is the transmitted intensity in mode i . The larger the PR, the more uniform the transmitted light is spread to all modes. For the measured transmission matrix at a fixed delay time, we calculate the PR for light injected to each fiber mode (each column) and average the PR over all input modes (all columns). To quantify the fluctuations of PR for different input modes, we compute the standard deviation of PR for all columns of the

transmission matrix. The mean value and the standard deviation of PR are plotted vs. the arrival time in a newly-added Fig. S1 of the Supplementary Materials. For comparison, we also compute the PR for isotropic random matrices. Each column is normalized to unity and consists of N complex elements that are randomly chosen from a uniform distribution. The ensemble-averaged PR is 0.51. The mean value of PR at different arrival times is close to 0.51 in Fig. S1. Hence, isotropy is a good approximation for the time-resolved transmission matrices of our fiber within the measurement range of delay times.

The slight deviations from the “isotropic” criteria are mainly caused by the mode dependent loss in the fiber. The higher order modes experience more loss than the lower order modes, thus having smaller magnitudes in transmission. Even without mode-dependent loss, the transmission matrices at very early (or late) arrival times tend to have more contributions from low-order (higher-order) modes with smaller (larger) group delays.

We have added a new subsection in the supplementary material, Section I. D, to discuss the isotropy of the transmission matrix.

2. The spatial C_2 correlation in scattering media has been with wavefront shaping (Vellekoop and Mosk, Universal Optimal Transmission of Light Through Disordered Materials PRL 2008). Upon focusing, the total transmitted energy is increasing via the C_2 correlations. If we try to find an analogy with the C_2 correlations in multimode fibre, would we see a similar effect? (i.e. the background at different times than t_0 should be enhance as well?). It seems not to be the case in Fig. 3c, or the effect is maybe too small to be observed. Hence it is not clear that the temporal shape of the achieved output pulse is only ruled by the measured so-called “spatio-temporal correlations”.

Vellekoop and Mosk studied spatial focusing of monochromatic light through scattering media. The enhanced total transmission they observed is due to the positive long-range correlation in space. For monochromatic light, multimode fibers exhibit *negative* long-range correlation in transmission, due to energy conservation (no reflection and negligible loss). When a monochromatic light is focused to one spatial channel of a multimode fiber, the intensities at other spatial channels must go down in order to preserve energy. Such difference highlights the qualitative difference between the long-range correlations in a multimode fiber and those in a diffusive random medium.

In this work, we consider broadband input (pulse) rather than monochromatic input (continuous wave), and study the enhancement of transmitted power (spatially integrated intensity) at certain arrival time. Naively, from the energy conservation, we would expect that when the transmitted power at a target arrival time increases, the transmitted power at all other arrival times should decrease for energy conservation. This is not exactly right, as our study shows. As shown in Fig. 3b, the transmitted power after the target time increases, while the transmitted power well before the target time decreases.

Such change is a direct result of $C_2(t, t')$, the long-range correlation of different arrival times. We added a curve of $C_2(t, t' = 16.1 \text{ ps})$ in Fig. 2e to explain Fig. 3b. The correlation is negative at early arrival time and positive at late arrival time. The negative correlation suppresses the background intensity and the positive correlation enhances the background. This is different from the study of Vellekoop and Mosk which shows only positive long-range correlation.

We cited the paper of Vellekoop and Mosk in the introduction, and added the above discussion in the left column on page 5.

3. In this paper the authors are only dealing with the long range correlations, and not about the C1 and the C3. I would suggest to modify slightly the title, which suggests the authors are studying more than the C2 effect. I would also recommend the author to cite this paper Amitonova et al., “Rotational memory effect of a multimode fiber”, Optics express 2015 where the rotational memory effect is demonstrated experimental in a MMF, as the angular memory effect cannot exist in a MMF.

Following the referee’s suggestion, we modified the title to “Long-range spatiotemporal correlations in multimode fibers for pulse delivery”. The paper “Rotational memory effect of a multimode fiber” studied the rotational memory effect, which is a result of short-range correlation C_1 . We cite this paper in the concluding paragraph of the revised manuscript.

4. In page 2 left column, the authors are stating “the correlations associated with an MMF is somewhat trivial, as $C_2 \sim 1/N$ is small”. Maybe the authors should explain a bit more, it might not be trivial to the non-expert reader.

Since there is no reflection in the fiber, the transmitted energy is conserved when the fiber loss is negligible. For a monochromatic light, the increase of transmitted intensity ΔI in mode a will cause a decrease ΔI in the sum of intensities of the rest $N - 1$ modes. Since these modes are statistically equivalent, on average the intensity in each mode will decrease by $\Delta I / (N - 1)$, which is approximately $\Delta I / N$ for $N \gg 1$. Therefore, for continuous wave input, the correlation between two spatial modes is $-1/N$ on average.

We have added the above explanation in the left column on page 2.

5. Regarding the experimental measurements, how long does it take to scan the wavelength for a fixed input state (end of page 4, right column)? Is the phase of the laser stable enough?

For each wavelength, we scan all input states to obtain the transmission matrix for this wavelength before moving to the next wavelength. It takes about 1 min to measure the field transmission matrix at a single wavelength. The phase stability of the laser does not have a significant influence on the measurement, because in our interferometric setup, the path-length of the fiber arm is matched to that of the reference arm. Even when the phase of the laser drifts, the relative phase between the two arms remains zero.

6. In page 3 left column, “considering a Gaussian transform-limited input pulse centered at wavelength 1550 nm with a full width at half maximum (FWHM) of 2.0 nm (temporal FWHM = 2.6 ps)”. How do you experimentally consider this when you measure your data, as you perform a monochromatic scan of the wavelength? You are not really sending an ultrashort pulse in your experiment. This point should be clarified. Why did you consider these numbers and not a broader source for instance? Is that limited by the resolution of your measurements?

We chose the input bandwidth to be much larger than the spectral correlation width of the multimode fiber. In our experiment, the fiber has a spectral correlation width of 0.2 nm, and the input bandwidth (FWHM) is equal to 10 times of the spectral correlation width. If a transform-limited pulse is sent to the fiber, the transmitted pulse will be 10 times broader than the input pulse. Namely, the pulse will experience strong temporal stretching due to modal dispersion, and the effect of temporal focusing by wavefront shaping is significant. In addition, the input

bandwidth is much larger than the spectral width of the principal mode (PM) or the super-PM, thus neither works for such broadband input.

To have a FWHM of 2.0 nm, we must scan a wider wavelength range to cover the tails, experimentally we scanned a wavelength range of 6.4 nm. With a wavelength step of 0.04 nm, we measured the field transmission matrices at 160 wavelengths. The entire measurement took about 2.5 hours. Of course, we could further increase the bandwidth, but at most by a factor of 2, otherwise the measurement would take too long, and the environment perturbations might destabilize the fiber.

We added the following sentence to the left column on page 3:

“The input bandwidth is 10 times of the spectral correlation width of the fiber, thus for random input wavefronts the transmitted pulse would be 10 times broader than the input pulse.”

The referee is correct that we were not sending a pulse to the fiber. Instead, after finding the maximal eigenchannel at the target arrival time, we generated the corresponding wavefront with a SLM for a CW laser beam. The wavelength of the CW laser was scanned, at each wavelength the transmitted field profile was recorded. The Fourier transform of the frequency-resolved field profiles gave the temporal shape of the transmitted field. This process was described in the original manuscript (right column, page 4) “By scanning the wavelength and Fourier transforming the spectral measurement to the time domain, we obtain the spatially integrated temporal pulse shapes of such optimal transmission eigenchannels”.

7. In Fig. 3b, why is the speckle size changing according to the delay time?

The transmission at early arrival time has more contribution from the lower order modes with smaller group delay, while the transmission at late arrival time has more contribution from higher order modes of larger group delay. Since the lower order mode have smaller speckles than the higher order mode, the speckle size is changing with the delay time.

We clarified this point by adding the following sentence to the caption of Fig. 3a-b:

“The speckle grains at $t_0 = -5$ ps are larger than those at $t_0 = 16.1$ ps, due to stronger contributions of the lower-order modes at earlier arrival time.”

8. How did you measure the error bars in Fig.3d? Did you try with the same fibre but different bending of the fibre? How many measurements did you perform to measure the error bars?

We performed 4 measurements of the same fiber under different bending configurations to obtain the error bar. The information is added to the caption of Fig. 3d.

9. In Fig. 4, what is defining and limiting the width of the temporal peak? Could it be narrower? (as you perform a spectral scan and you don't have any limitation of the time width of an ultrashort pulse) How do you normalize the intensities in Fig.4a and Fig.4b? How do you measure the speckle with the spatial resolution at a chosen time? Is there a reason why light is mostly focused on two speckle grains in Fig. 4a and Fig.4b when you enhance pulse delivery at a chosen delay?

The width of the temporal peak is given by the temporal speckle size, which is inversely proportional to the spectral bandwidth of the input pulse. The peak width is equal to the input pulse width if the input pulse is transform-limited, as considered in the paper. The temporal peak

could be narrower if the input bandwidth was broader. The reason that we set the input bandwidth to 2nm is given in the answer 6. The spatial field profile at the target arrival time was obtained from the frequency-resolved measurement, as explained in answer 6.

To clarify these two points, we added the following explanation in the left column, page 5.

“The peak width equals that of the input pulse. The spatial intensity patterns of the optimized pulse and the non-optimized one at the target arrival times, shown in Fig. 3a-b, are obtained from the Fourier transform of the frequency-resolved field patterns measured with optimal and random incident wavefronts.”

We believe the reviewer asked about the normalization in Fig. 3a and 3b, as there is no Fig. 4. In the original manuscript, the curves are normalized so that the total energy of transmitted pulse for random input wavefront is equal to unity. We realize that this normalization may be confusing, in the revised manuscript, the peak power of transmitted pulse for random incident wavefront is normalized to one in Fig. 3a and 3b. We added the following sentence to the caption of Fig. 3:

“Transmitted powers are all normalized by the peak power of the transmitted pulse for random input wavefronts.”

Regarding the two bright speckle grains in Fig. 3a and Fig. 3b, we calculated the power they carry and found only 28% (Fig. 3a) or 16% (Fig. 3b) of the total power is in these two speckle grains. Most of the energy is still distributed over other speckle grains. Because the probability density function (PDF) of speckle intensity features an exponential decay (Rayleigh statistics), bright speckle grains are fewer than dim speckle grains. The speckle pattern of the random input wavefront in Fig. 3a also shows two bright speckle grains. Hence, they are not a consequence of enhanced pulse delivery at a chosen delay.

10. In page 4 left column “the large $C_2(t)$ at early and late arrival times is consistent with the lower number of paths at such times”, maybe the author should plot the number of modes at such times, to make it more clear? Could this be quantitative?

We would like to clarify that the number of paths and the number of modes are two different quantities. Due to mode coupling, light will hop from one mode to another while propagating in the fiber. There are many possible paths in a multimode fiber. Even though the number of modes in the fiber is fixed, the number of paths increases with the fiber length and the path-length distribution broadens.

By varying the incident wavefronts, all paths in the fiber are explored. The number of paths with identical length is proportional to the transmitted light intensity at the corresponding transit time. As shown in Fig.1b, the transmitted pulse shape, averaged over many random incident wavefronts, gives the path-length distribution in the fiber.

The above discussion is added to page 3, right column.

11. This sentence in page 5 left column “The background change is a result of long-range spatio-temporal correlations $C_2(t; t_0)$. Namely, the negative correlations for well-separated times t and t_0 ($t_0 = t_0$) shift the background toward the target time t_0 ” is not clear

In our reply to Question 2, we have explained how C_2 correlation determines the optimized pulse shape. Because $C_2(t, t_0)$ (shown in Fig. 2d) is negative for the arrival time t far from the target

time t_0 and becomes positive for t close to t_0 , the transmitted power is enhanced close to the target time and suppressed far from the target time. Consequently, the background pulse moves toward the peak at the target time t_0 .

We elaborate this point on page 5, right column.

12. In page 5 left column, “The average enhancement is about 4 times around the central arrival time, which is what one expects through the quarter-circle law for the singular values of a square random matrix with uncorrelated elements”. Don’t you have any ballistic light in multimode fibres, even in the strong mode mixing regime? An evidence of this would be highly recommended.

If there were ballistic light, the diagonal elements of the transmission matrix would be stronger than the non-diagonal elements. This is not the case for the measured transmission matrices shown in Fig. 1c-e. Therefore, the ballistic light is negligible in our fiber.

To clarify this point, we add the following sentence to page 3, left column

“The absence of a dominant diagonal reveals negligible ballistic light at fiber output.”

13. Are the measured C_2 correlations the only parameters to help building up the pulse delivery? In the case of the spatio-temporal focusing in multimode fibre (Morales-Delgado et al. Optics Express Vol. 23, Issue 7, pp. 9109-9120 (2015)), should the C_2 correlation play a role as well in the digital phase conjugation results? If you were not in a strong mode-mixing regime, would you expect similar results, slightly better or worse?

With strong mode mixing in a multimode fiber, the long-range dynamic correlation C_2 is the only parameter that determines the pulse delivery. Eq. (4) shows that the enhancement of transmitted power at the target time is determined by $C_2(t_0) = C_2(t_0, t_0)$ and the number of fiber modes N . The temporal pulse shape is predicted by $C_2(t, t_0)$ from Eq. S8. C_2 correlation also determines the peak-to-background contrast of spatio-temporal focusing in a multimode fiber.

If the fiber is not in the strong mode coupling regime, the field transmission matrices are highly anisotropic, and the results obtained here no longer hold. Nevertheless, the long-range correlation is still strong and plays a significant role in temporal focusing in the multimode fiber. We numerically calculate $C_2(t_0)$ for strong mode coupling and weakened mode coupling in the same multimode fiber and plot them in the figure below. Further study is needed for the multimode fiber with weak mode coupling, which is beyond the scope of the current study.

Fig. R2. Time-dependence of the long-range correlation $C_2(t_0)$ in multimode fibers with strong mode coupling (red solid curve) and weakened mode coupling. The arrival time is normalized by the input pulse width. In both cases, the spectral bandwidth of the input pulse is set to 10 times of the spectral correlation width.

14. In page 5 right column. “For the broad-band pulses studied here”, this sentence is incorrect, as you are not really sending a short pulse in the experimental results.

The referee is correct that we are not sending an optical pulse to the fiber, instead we conducted the experiment in frequency domain and converted the results to the time domain. To be clear, we modified this sentence in the revised manuscript.

15. In the Supplementary information part C, what is the delay Δt over which $C_2(\Delta t)$ goes to 0?

The reviewer is likely referring to the sentence “When t is far from t_0 , the correlation $C_2(t, t_0) \approx 0$ ” in the Supplementary Materials. From Fig. 2d, we see that the delay $\Delta t = t - t_0$ for which $C_2(t, t_0)$ goes to zero depends on the target time t_0 . Once $C_2(t, t_0)$ is 0, the intensity at time t should be equal to that of a random input and the enhancement should be 1.

For clarification, we changed this sentence to “when the correlation $C_2(t, t_0) \approx 0$, the power at t should be equal to that of random incident wavefront, giving $\eta(t, t_0) \approx 1$ ”.

16. My final question is about your comparison between the achieved state and the super principal modes. Is there a physical relationship between the principal modes and the EM? For instance, the bandwidth in the spectral correlation when you calculate the principal modes is at 0.9. If you would like to optimize the bandwidth at 0.6 (which seems to be where the bandwidth of the EM modes are the broadest) instead of 0.9, could you generate an equivalent of the EM based on the principal modes measurement only?

The principal modes (PMs) are obtained from the eigenstates of the time-delay operator (group-delay matrix), not from optimizing the bandwidth. Thus defining their bandwidth at 0.6 or 0.9 would not alter the PMs, or the temporal shape of transmitted pulse. The super-PMs are obtained from minimizing the spectral decorrelation within a frequency range, i.e., minimizing the total area underneath the curve of the spectral correlation function, and again defining their bandwidth at 0.6 or 0.9 would not matter. To compare PM, super-PM and EM (eigenmodes of the time-resolved field transmission matrix), we vary the spectral bandwidth of incident pulse, and plot the temporal shape of transmitted pulse in Fig. S2.

When the input bandwidth is 10 times of the spectral correlation width of the fiber, EM clearly outperforms PM and SPM in both the frequency domain and the time domain (a,b). EM achieves the highest peak power of the transmitted pulse. When the input bandwidth is reduced to 5 times of the spectral correlation width (c,d), the PM still displays notable spectral decorrelation and the output pulse is notably lengthened in time. The SPM and EM are effective in suppressing spectral decorrelation and temporal stretching, but EM still outperforms SPM slightly. Only when the input spectrum is very narrow (e,f), which is set to twice of the spectral correlation width of the fiber, PM, SPM and EM achieve almost equivalent performances.

The above discussions are now added to Section III of the Supplementary Materials.

Side remarks:

17. In Fig. 1b, there must be a problem of unit of the input power if the energy is conserved (the max must be 0.4 and not 0.04?)

We thank the reviewer for pointing out the typo in Fig. 1b. Indeed, the max is 0.4 instead of 0.04. In the revised figure, we normalize both the input and the output pulse by the maximum of the output pulse.

18. What is the physical meaning of $N^{\text{eff}}(t_0)$?

In a random matrix with correlated matrix elements, the size of the matrix does not capture the effective degrees of freedom, because the different output channels will change in a correlated manner as the input is varied. The $N^{\text{eff}}(t_0)$ we define captures the effective number of output channels—the number of independent degrees of freedom at the output.

If the authors could clarify those points it would be highly appreciated.

We thank the referee again for a thorough and in-depth review, and we have clarified all the points raised by the referee in the revised manuscript.

Reviewers' comments:

Reviewer #2 (Remarks to the Author):

The authors have satisfactorily answered all questions raised by the reviewers and modified the manuscript accordingly. This has clarified their work on pulsed transmission in MMFs and, at the same time, highlighted the importance of correlation in controlling transmission in any system. I therefore strongly recommend that the paper should be accepted for publication in Nature communications in its present form.

Reviewer #3 (Remarks to the Author):

The authors replied with details to all the questions. I appreciate the discussion on the isotropy of the transmission matrix, which was clearly missing in the previous version of the manuscript. I only have a couple of remarks:

1. The authors are not really explaining the physical origin of this effect (e.g. what are the concepts behind this effect, how can we interpret it with a hand waving reasoning, can we physically adjust/tune this effect, etc...).
2. Why is the long range correlation effect not present (e.g. $C_2=0$) for middle delay arrival times (between -5ps and 10ps according to Fig. 2d)? Would it mean that with a fiber that supports a higher number of modes, the C_2 effect would be negligible?

Side remarks:

1. The colormap has not be updated in Fig. 3c, the values are with the previous normalization the authors were using.
2. I don't know if normalizing the plots in Fig. 3 to the maximum of the random incident wavefront makes more sense than normalizing its integral. Normalizing by the integral would actually make more sense as the output should carry the same total power whatever the input is, as there is almost no loss in this fiber.

The results of this manuscript are interesting and original, as the study of mesoscopic correlations is mostly theoretical, and hardly comes with experimental results. However, I am not sure if these results would meet the criteria of a general high impact journal rather than a more specialized journal.

Reviewer #3 (Remarks to the Author):

We thank Reviewer #3 for raising some additional points, which we address below. We have also revised the paper based on these remarks.

1. The authors are not really explaining the physical origin of this effect (e.g. what are the concepts behind this effect, how can we interpret it with a hand waving reasoning, can we physically adjust/tune this effect, etc...).

We did explain the physical origin for the long-range spatio-temporal correlation in the paper. In the last paragraph on page 3, we provided a qualitative explanation about the time dependence of $C_2(t)$ using the optical path-length distribution in the multimode fiber:

“The large $C_2(t)$ at early and late arrival times is consistent with the lower number of paths for such times. Conceptually, if there is only one path of length corresponding to the arrival time t , the output intensities $I(r,t)$ at different positions r must be fully correlated: varying the incident wavefront can only change how much light is coupled into that path, which will increase or decrease $I(r,t)$ at all positions in the same way. Therefore, the fewer paths for the arrival time t , the stronger $C_2(t)$.”

To answer Reviewer #3’s question about physically adjust/tune this effect, we added the following sentence to the last paragraph on page 5:

“We provide a qualitative explanation for the long-range spatio-temporal correlations using the optical path-length distribution in the multimode fiber with random mode mixing. This simple model reveals the possibility of physically tuning the spatio-temporal correlations by tailoring the path-length distribution in the fiber via a careful design of the fiber configuration.”

2. Why is the long range correlation effect not present (e.g. $C_2=0$) for middle delay arrival times (between -5ps and 10ps according to Fig. 2d)? Would it mean that with a fiber that supports a higher number of modes, the C_2 effect would be negligible?

Based on the path-length explanation, the negligible long-range correlation for middle delay arrival times is due to the existence of a large number of paths for such arrival times. As we mentioned in page 3, “... the averaged temporal shape of the transmitted pulse in Fig. 1b reflects the number of propagation paths with varying lengths”. At the middle delay time, the temporal pulse reaches the maximum intensity, so the number of available paths is the largest and the long-range correlation effect is the weakest. To explain this more clearly, we added the following sentence to the last paragraph in page 3:

“For the middle delay times, there are a large number of propagation paths, thus $C_2(t)$ is very weak.”

Indeed, the higher the number of modes in the fiber, the weaker the C_2 effect. In page 2, we showed that the static long-range correlation, which corresponds to a continuous wave input, is negligible, $\tilde{C}_2 \approx -1/N$, when the number of fiber modes $N \gg 1$. But for pulsed inputs, C_2 effect becomes non-negligible at very early or late arrival times, even for the fiber with $N \gg 1$, as demonstrated in our experiments.

Side remarks:

1. The colormap has not be updated in Fig. 3c, the values are with the previous normalization the authors were using.

The colorbar of Fig. 3c is now corrected.

2. I don't know if normalizing the plots in Fig. 3 to the maximum of the random incident wavefront makes more sense than normalizing its integral. Normalizing by the integral would actually make more sense as the output should carry the same total power whatever the input is, as there is almost no loss in this fiber.

The choice of normalization is a matter of presentation. In the original manuscript, we normalized the integral in Fig. 3, as Reviewer #3 suggested now. But Reveiwer#3 asked about the normalization in the previous report, which made us realize that such normalization might be confusing. Thus we normalized the peak power instead, as we felt such normalization would be clearer and easier to understand. We do not think that the normalization by the integrated intensity is more meaningful. Because the fiber used in the experiment has the mode-dependent loss, pulses with enhanced intensity at later arrival times has less total energy due to longer path-length in the fiber. Therefore, the pulse energy is not conserved, when the pulse is optimized for different arrival time.

The results of this manuscript are interesting and original, as the study of mesoscopic correlations is mostly theoretical, and hardly comes with experimental results. However, I am not sure if these results would meet the criteria of a general high impact journal rather than a more specialized journal.

We respectfully disagree that our paper does not meet the criterial of Nature Communications. Nature Communications publish papers representing "important advances of significance to specialists within each field" (quoted from the official website). Our study of spatio-temporal long-range correlations in multimode fibers is an important discovery and represents a significant advance that is of general interest to both mesoscopic physics and optical fiber communities.

REVIEWERS' COMMENTS:

Reviewer #3 (Remarks to the Author):

The report of the authors is convincing. The new manuscript has been improved since first submission, and I suggest publication in Nature Communications.